# MerKury: Adaptive Resource Allocation to Enhance the Kubernetes Performance for Large-Scale Clusters

## ABSTRACT

As a prevalent paradigm of modern web applications, cloud computing has experienced a surge in adoption. The deployment of vast and various workloads encapsulated within containers has become ubiquitous across cloud platforms, imposing substantial demands on the supporting infrastructure. However, Kubernetes (k8s), the de-facto standard for container orchestration, struggles with low scheduling throughput and high latency in large-scale clusters. The primary challenges are identified as excessive loads of read requests and resource contention among co-located components.

In response to these challenges, in this paper, we present MerKury, a lightweight framework to enhance the Kubernetes performance for large-scale clusters. It employs a dual strategy: first, it preprocesses specific requests to alleviate unnecessary load, and second, it introduces an adaptive resource allocation algorithm to mitigate resource contention. Evaluations under different scenarios of varying cluster scale have demonstrated that MerKury notably augments cluster scheduling throughput up to 16.4× and reduces request latency by up to 39.3%, outperforming vanilla Kubernetes and baseline resource allocation methods.

## CCS CONCEPTS

• **Computer systems organization → Cloud computing**.

## KEYWORDS

resource allocation, Kubernetes, large-scale

**ACM Reference Format:**

Anonymous Author(s). 2024. MerKury: Adaptive Resource Allocation to Enhance the Kubernetes Performance for Large-Scale Clusters. In *Proceedings of The Web Conference 2025 (WWW '25)*. ACM, New York, NY, USA, 12 pages. https://doi.org/10.1145/nnnnnnn.nnnnnnn

## 1 INTRODUCTION

Cloud computing has witnessed exponential growth, establishing itself as a preeminent computing paradigm for web applications. The proliferation of diverse workloads, including microservices [6, 29], batch processing jobs [8, 23], and Function as a Service (FaaS) [30, 34], has led to a significant expansion in the scale of nodes and containers on cloud platforms. This, in turn, exerts considerable pressure on the underlying infrastructure.

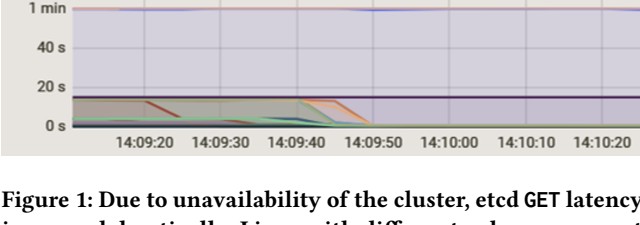

**Figure 1: Due to unavailability of the cluster, etcd `GET` latency increased drastically. Lines with different colors represent different types of requests, among which the P99 latency of `GET` pod (the light brown line) exceeded one minute.**

Kubernetes (k8s), the industry's de facto standard for container orchestration, is recognized as a pivotal part of cloud infrastructure [16]. However, it exhibits limitations in scaling and encounters multi-dimensional constraints. For instance, a cluster is restricted from surpassing 5,000 nodes, and the pod count per node is capped at 110 [18]. In large-scale clusters, Kubernetes performance degrades severely when dealing with an overwhelming influx of requests. Initially, the request latency escalates. The API server experiences prolonged response times [7, 27], and the scheduler exhibits diminished throughput during pod scheduling operations [7, 35]. Moreover, cluster availability is compromised. As depicted in Fig. 1, in a cluster with 2,000 nodes and 60,000 pods, an unexpected network outage triggered a surge in pod reconnection requests. This surge overwhelmed the control plane's capacity, precipitating repeated crashes and restarts due to OOM (Out-Of-Memory) issues, leading to the unavailability of the entire cluster.

To tackle Kubernetes' scalability issues and facilitate the unified management of large-scale workloads, two main strategies have emerged. Multi-cluster strategies utilize cluster federation technologies [1] to manage multiple clusters as a hyperscale entity [13], while single-cluster strategies improve cluster capacity through optimizing core components [5, 10, 35] and mechanisms [7, 36]. However, the management layer introduced by multi-cluster strategies can cause additional complexity and resource overheads [24]. Moreover, most studies cater to specific scenarios, such as far edge nodes [36], and change the Kubernetes codebase, which may hinder their broader application.

In this paper, we introduce MerKury, a general and lightweight framework designed to enhance the Kubernetes performance for large-scale clusters. Motivated by the inefficiencies in read request processing and the resource contention among components co-located on the master node, MerKury preprocesses requests to alleviate unnecessary load, and dynamically adjusts the resource allocation and traffic control parameters of control plane components to mitigate resource contention. We seamlessly integrate MerKury as a non-intrusive plugin, requiring no alterations to Kubernetes

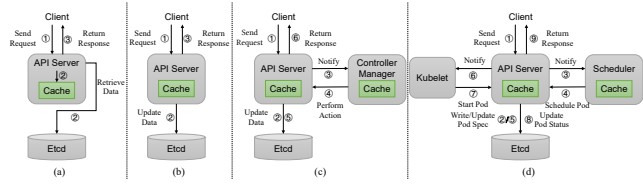

**Figure 2: Kubernetes request processing workflows of (a) read requests, (b) simple write requests without state reconciliation, (c) write requests involving the controller, and (d) pod creation or update requests.**

codebase. Our comprehensive evaluation across various cluster sizes and request volumes has demonstrated MerKury's substantial performance gains. Notably, it outperforms vanilla Kubernetes and baseline resource allocation strategies by enhancing scheduling throughput by up to 16.4 times and reducing request latency by up to 39.3%. Additionally, the lightweight design ensures minimal resource and time overhead, with negligible impact on business workloads.

The contributions of this paper are threefold. Firstly, we analyze typical Kubernetes requests and introduce streamlined preprocessing methods to alleviate unnecessary load on control plane components. Secondly, we develop a queuing model for request processing and design an algorithm for the adaptive resource allocation and traffic control parameters tuning, which significantly mitigates resource contention. Lastly, our non-intrusive implementation of MerKury has been rigorously tested and demonstrated superior performance compared to vanilla Kubernetes and other resource allocation strategies. The code of MerKury is available on GitHub[1].

## 2 BACKGROUND AND MOTIVATION

### 2.1 Kubernetes Request Processing Flow

In Kubernetes, control plane components–the API server, scheduler, controller manager, and etcd–collaborate to process requests initiated by users and the system. We categorize them into four types based on the components they interact with. Read requests primarily engage the API server, with etcd sometimes participating. Simple write requests, which do not require state reconciliation, involve both the API server and etcd. Stateful resources writes additionally incorporate the controller manager. Lastly, pod creations and updates involve a coordinated effort between the API server, etcd, scheduler, and the kubelet on worker nodes. The distinct processing workflows are illustrated in Fig. 2.

### 2.2 Problems

While control plane components are adept at maintaining robust operations in small to medium-sized clusters, their performance in large-scale environments is notably diminished. Our tests, detailed in § 5, quantify the performance degradation, revealing a significant rise request latency, especially for read requests, and a maximum 38.5% drop in scheduling throughput as the cluster size grows from 1,000 to 5,000 nodes as shown in Table 1. Monitoring and analyzing

[1]https://anonymous.4open.science/r/merkury-www-C673

**Table 1: Performance of vanilla Kubernetes (k8s-static).**

| Number of nodes | Average latency of all requests (ms) | P99 latency of read requests (ms) | Scheduling throughput (pod/s) |
|---|---|---|---|
| 1,000 | 45.2 | 1200.1 | 42.0 |
| 2,000 | 100.3 | 3418.7 | 43.6 |
| 3,000 | 129.9 | 8252.9 | 36.4 |
| 4,000 | 115.9 | 7244.0 | 31.0 |
| 5,000 | 137.1 | 9141.2 | 26.8 |

cluster metrics, we have identified two key performance detractors.

*2.2.1 Excessive Loads of Read Requests.* Read requests, including GET and LIST operations, are pivotal to Kubernetes performance. However, suboptimal configurations can lead to excessive loads. Typically, the API server can source data from the local cache or etcd, with the latter introducing a much higher load due to overhead of data transmission and the absence of filtering mechanisms. In extreme scenarios, a LIST request can generate a load magnitudes higher when data is retrieved from etcd instead of the local cache.

*2.2.2 Resource Contention Among Co-located Components.* On Kubernetes master nodes, concurrent operation of multiple control plane components can lead to resource contention, especially under heavy load. CPU contention can cause performance degradation and latency. Memory contention can lead to OOM errors and component crashes, impacting control plane availability. The default Kubernetes configuration does not consider resource allocation, and in a high-availability setup, the arbitrary placement of master components can concentrate them on a single node, exacerbating resource contention and performance decline.

## 3 RELATED WORK

### 3.1 Kubernetes Optimization

Literature on Kubernetes optimization focus on enhancing the control plane's scalability and reliability, as well as the data plane's efficiency. For the control plane, KOLE [36] improves scalability through MQTT messaging, while Gödel [35] increases scheduling throughput with a parallel framework. Tools like Sieve [32] and Acto [11] bolster reliability by detecting issues in controllers and operators. On the data plane, AHPA [37] conserves pod resources and maintains business stability, while Optum [25] improves the overall resource utilization.

MerKury distinguishes itself by enhancing the control plane in general purpose without altering the Kubernetes codebase, ensuring broad applicability and ease of deployment.

### 3.2 Resource Allocation

Resource allocation is critical in cloud computing, influencing fairness, cost, and performance. Karma [33] uses a credit-based algorithm for equitable sharing, while StepConf [34] automates resource configuration for cost-effective serverless functions. Queuing theory is prevalent in formulating allocation problems [4, 12, 21], and heuristic algorithms are favored in solutions for their effectiveness in dynamic cloud scenarios [2–4].

MerKury distinguishes itself by introducing an innovative queuing model with dynamic CPU-concurrency mapping, modelling

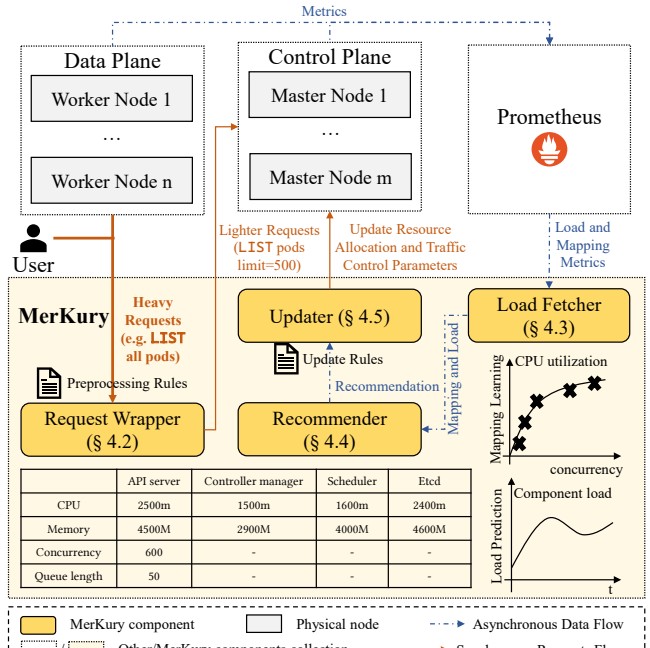

**Figure 3: MerKury's architecture highlighting the main components in yellow. The data flow is centered on optimizing resource allocation and traffic control, while the request flow includes preprocessing and optimization implementation.**

request handling more precisely. In addition, it utilizes readily available metrics and accounts for component dependencies, balancing between the simplicity and precision of problem formulation. Moreover, the heuristic algorithm offers an efficient solution.

## 4 MERKURY DESIGN

This section introduces the design of MerKury. We first present the design principles and overview the architecture, and subsequent sections elaborates on its four main components.

### 4.1 System Overview

Understanding the foundational design principles of MerKury is crucial before delving into its details. MerKury is characterized by its **broad applicability**, setting it apart from previous systems that were tailored to special use cases [35, 36]. It is designed for general scenarios and integrates seamlessly with Kubernetes. Furthermore, MerKury is engineered for **efficiency and lightweightness**, enhancing cluster performance with acceptable resource and time cost, ensuring it does not become a burden on the system.

Fig. 3 illustrates MerKury's architecture. Requests from users and the data plane are first directed to the **request wrapper** (§ 4.2), which preprocesses them in real time to alleviate excessive load. For each control plane component, the **load fetcher** (§ 4.3) retrieves load and mapping metrics from Prometheus to evaluate its load and update its CPU-concurrency mapping. For each master node, the **recommender** (§ 4.4) periodically creates recommendations for CPU allocation and traffic control parameters based

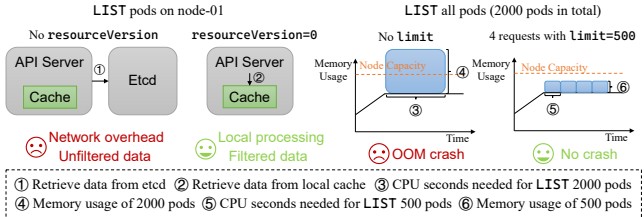

**Figure 4: Preprocessing Rules.** The *left* panel shows that by setting `resourceVersion=0`, local data filtering in the API server eliminates the network overhead. The *right* panel depicts the segmentation of large `LIST` requests into smaller chunks, averting potential out-of-memory issues.

on data from the load fetcher. The **updater** (§ 4.5) translates valid recommendations into executable requests sent to the control plane, implementing recommendations and master instance placement.

### 4.2 Request Wrapper

As highlighted in § 2.2.1, read requests are pivotal to Kubernetes performance, and their misconfiguration can impose a significant burden. The request wrapper is designed to tackle this issue by implementing request preprocessing rules, depicted in Fig. 4.

**Minimize Etcd Access for Read Requests.** Although etcd maintains the latest object data, its access can be onerous due to network latency and its lack of in-built filtering capabilities. The `resourceVersion` parameter is the determinant, whose absence triggers a data retrieval from etcd. To minimize etcd access, MerKury automatically appends `resourceVersion=0` to read requests missing this parameter, avoiding direct etcd queries and leveraging the API server's local cache instead.

**Cap the Size of Objects Returned by Read Requests.** `LIST` operations in large-scale clusters can generate massive responses, whose large data volumes can severely impact the cluster's performance and availability, particularly under heavy load. To mitigate this, MerKury caps requests at `limit=500`, breaking down heavy requests into lighter ones to keep data volume manageable and curb excessive memory use.

The request wrapper applies these rules to external requests in real-time, thus preserving the integrity of internal control plane operations and avoiding unintended disruptions.

### 4.3 Load Fetcher

The load fetcher acquires and processes load metrics and mapping metrics vital for the recommender.

*4.3.1 Load Metrics.* Load metrics, reflecting resource consumption and request intensity such as CPU, memory utilization, and RPS, are crucial for performance modelling. Unlike Kubernetes VPA [19], which focuses solely on resource usage, or previous studies [2, 21] utilizing complex metrics that requires extensive efforts to obtain, MerKury's load fetcher prioritizes easily accessible metrics. It balances model accuracy with development simplicity by selecting load metrics including incremental indicators (processed requests, CPU time slices, allocated memory) and status indicators (queuing requests, CPU and memory utilization, CPU throttling percentage).

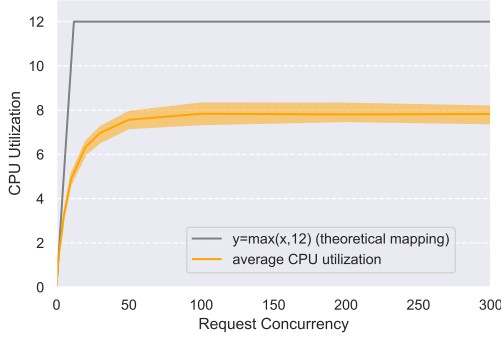

Figure 5: CPU-concurrency mapping comparison between theoretical (M/M/c model) and actual API server performance. Data from repeated stress tests for LIST configmaps requests.

*4.3.2 Mapping Metrics.* previous studies [4, 22] often use basic queuing models such as M/M/1 and M/M/c [31] to model the system performance. These models assume a linear relationship that does not reflect the complex reality of CPU-concurrency mapping. To capture the mapping accurately, we have conducted a series of stress test using wrk to simulate different load conditions on an API server with a 12-core CPU. The results, shown in Fig. 5, revealed that the mapping is *nonlinear* and *dynamic*. Therefore, the load fetcher periodically aggregates recent (concurrency, CPU utilization) data points and derives the mapping $f(x)$ through interpolation. Mapping metrics are collected solely for the API server due to the unavailability of other components' concurrency metrics.

## 4.4 Recommender

The recommender provides recommendations for resource allocation and traffic control parameters to alleviate resource contention.

*4.4.1 Queuing Model.* We model the request processing of a control plane component as a queuing system where CPU cores act as servers and requests as customers. Requests alternate between queuing and execution states. The queuing state incurs memory load $m_q$ to store requests and their context, while the execution phase involves CPU load $c$ for processing and memory load $m_e$ for intermediate data storage. Within a time frame $\Delta t$, $n_r$ requests with load $\vec{L} = (m_q, c, m_e)$ arrive. The component's CPU allocation is $c^*$, with maximum concurrency $f^*$ and queue length $q^*$.

The system can be characterized by a birth-death process as shown in Fig. 6. When there are $n$ requests in the system, the arrival rate $\lambda_n$ and service rate and $\mu_n$ are:

$$\lambda_n = \lambda \quad (0 \le n < f^* + q^*),$$

$$\mu_n = \begin{cases} \min\left(c^*, f(n)\right)\mu, & (1 \le n < f^*) \\ \min\left(c^*, f(f^*)\right)\mu. & (f^* \le n \le f^* + q^*) \end{cases} \quad (1)$$

Here, $\lambda = n_r/\Delta t$ is the average inter-arrival time inverse, and $\mu = c/n_r$ is the average request processing time inverse.

In the steady state when the service rate exceeds the arrival rate, the probability distribution $\{p_n\}$ is derived by solving flow balance equations [31]. The distribution allows us to calculate performance

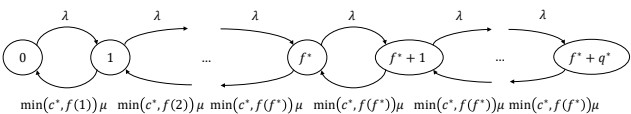

Figure 6: Rate transition diagram for the birth-death process.

metrics such as the expected numbers of total $L$, queued $L_q$, and executing $L_e$ requests. Using Little's Law, we determine the average request latency $W$, queuing $W_q$ and execution time $W_e$.

In the unsteady state when the arrival rate exceeds the service rate, the unprocessed CPU load, denoted as $l$, is estimated as:

$$l = \max\left(0, c - \mu\Delta t\right) = \max\left(0, c - \min\left(c^*, f(f^*)\right)\Delta t\right). \quad (2)$$

Regardless of the steadiness, memory usage can be estimated as follows, taking into account the baseline memory consumption $m_0$:

$$m = m_0 + \frac{L_q m_q}{n_r} + \frac{L_e m_e}{n_r}. \quad (3)$$

*4.4.2 Problem Formalization and Solution.* With the queuing model established, it is intuitive to formalize the problem as minimizing the weighted average request latency $\bar{W}$. However, this formalization has overlooked important factors including differences and dependencies among components, and steady-state conditions.

**Components Classification and Dependencies.** We categorize components into three groups based on their SLOs [17].

*Group A: API Server and Etcd.* The goal is to minimize their weighted average request latency.

*Group B: Scheduler.* The aim is to maximize the scheduling throughput, $S$, calculated over the last time frame for $n_r$ pods with total CPU load $c$, and $n_q$ pending pods:

$$S = \min\left(\frac{n_q}{\Delta t}, \frac{c^* n_r}{c}\right). \quad (4)$$

*Group C: Controller Manager.* The target is to allocate sufficient CPU for prompt state reconciliation of objects in the work queue. If the predicted CPU load for the next time frame is $c'$, the CPU allocation for the controller manager is given by:

$$c^* = \max\left(1, \frac{w_C}{\bar{w}}\right) c'. \quad (5)$$

Here, $w_C$ and $\bar{w}$ represent allocation weights discussed in § 4.4.3.

As shown in Fig. 2, there are dependencies among components. For instance, more CPU allocation for scheduler and controller manager may not only speed up their RPS, but also increase the load for API server and etcd. For group A components, requests are from outside $n_{r,o}$, group B $n_{r,B}$, and group C $n_{r,C}$. Based on the CPU allocation for groups B $c_B^*$ and C $c_C^*$, for the next time frame, the requests originating from these groups should be calibrated as:

$$n'_{r,B} = \frac{S\Delta t}{n_{r,B}} n_{r,B}, \quad n'_{r,C} = \frac{c_C^*}{c_C} n_{r,C}. \quad (6)$$

**Steady-state Conditions and Constraints.** The scheduler operates asynchronously, processing pod scheduling requests without requiring immediate responses, making the steady-state concept inapplicable. For other components, the steady-state condition is the service rate exceeding the arrival rate:

$$\min\left(c^*, f(f^*)\right)\frac{n_r}{c} > \frac{n_r}{\Delta t} \implies \min\left(c^*, f(f^*)\right) > \frac{c}{\Delta t}. \quad (7)$$

There are universal constraints that apply regardless of steady-state conditions. Specifically, the total allocated CPU resources must not surpass the master node's capacity, and the cluster's availability must meet the Service Level Objective (SLO):

$$\sum c^* \leq C, \quad 1 - p_{\text{reject}} \geq P. \tag{8}$$

Here, $C$ represent the total CPU resources of the master node, $p_{\text{reject}}$ denote the probability of a request being rejected, and $P$ is the minimum availability threshold.

**Steady State Problem.** The objectives in the steady state are to maximize scheduling throughput and minimize the weighted average latency for group A components, formalized as:

$$\min \quad w_A \bar{W} + w_B \frac{1}{S+1}, \tag{9}$$
$$\text{s.t.} \quad (7)(8).$$

Here, $w_A$ and $w_B$ are the weights assigned to balance the optimization objectives for different components.

To solve (9), one can employ standard optimization techniques such as evolutionary algorithms. Rather than resorting to a brute force approach that exhaustively searches the entire solution space, we seek heuristic algorithms to improve the efficiency.

To maximize scheduling throughput, under the constraint (7), $c^*$ should be maximized within the limit of $n_q c / n_r \Delta t$ according to (4).

Minimizing latency is more complex and requires an analysis of the monotonicity of $W$ with respect to $c^*$, $f^*$, and $q^*$. Theoretical proofs indicate that: ① $W$ decreases as $c^*$ increases; ② the monotonicity between $W$ and $f^*$ is inconclusive, but $L$ increases with respect to $f^*$ if increasing $f^*$ does not enhance the service rate; ③ the monotonicity between $W$ and $q^*$ is inconclusive, but $p_{\text{reject}}$ decreases with $q^*$, while $L$, $L_e$, and $L_q$ increase with $q^*$. Based on these conclusions, the *allocation method for parameters* $(c^*, f^*, q^*)$ is derived. Initially, $c^*$ should be maximized within the limit of $\max f(x)$ to avoid allocation waste. After determining $c^*$, identify the intersection points where $f(x) = c^*$ and select the smallest x-coordinate as $f^*$ to ensure full CPU utilization without increasing $L$. Finally, under the constraint of cluster's availability, choose the smallest $q^*$ to mitigate memory pressure.

**Unsteady State Problem.** The steady-state condition may be violated in two scenarios. These scenarios can occur simultaneously and can have a detrimental impact on components, with the severity increasing as the time to process the remaining CPU load lengthens.

The first scenario, termed *global-unsteady state*, occurs when CPU allocation is insufficient for immediate request processing. In the global-unsteady state, the objective is to minimize the adverse effects on components other than the scheduler, and to balance the negative impacts among them, preventing any single component from becoming a severe bottleneck that could affect the overall performance and availability of the cluster. Thus, the optimization problem in the global unsteady state is formulated as:

$$\min \quad \sum_i l_i, \tag{10}$$
$$\text{s.t.} \quad t = l_i / c_i^*, \quad (8).$$

Here, $t$ represents the time required to process the remaining CPU load. According to (2), $l$ decreases with $c^*$, and hence the objective is achieved when $\sum_{i=1}^{n} c_i^* = C$. Therefore, the solution for (10)

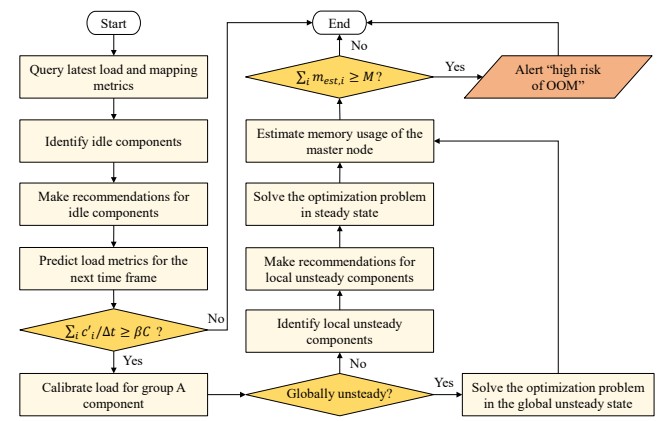

**Figure 7: Flow chart of the recommendation algorithm.**

is $c_i^* = \frac{c_i' C}{\sum_i c_i}$, i.e. CPU resources are allocated proportionally to the calibrated CPU load of each component. For traffic control parameters, the selection of $f^*$ is the same as in the steady state, while $q^* = L_q$ to accommodate all requests.

The second scenario, termed *local-unsteady state*, is characterized by the insufficient maximum CPU utilization even when sufficient CPU resources are allocated, as shown in Fig. 5. In the local-unsteady state, $c^* > \max f(x)$ could result in waste. However, given the dynamic nature of $f(x)$, a component might utilize CPU more than $\max f(x)$ and potentially return to a steady state. Therefore, when there are sufficient allocatable CPU resources, we adopt an *optimistic strategy*. This strategy allocates $c / \Delta t$ CPU resources to meet the steady-state conditions regardless of $\max f(x)$.

*4.4.3 Recommender's Workflow.* The recommender operates periodically, following the workflow depicted in Fig. 7.

Each cycle consists of the following steps: ① *Data Collection.* The recommender first queries the latest load and mapping metrics from load fetchers. ② *Idle Components Identification.* For group A components, an idle status is assigned if there is not any processed or queuing requests. For other components, it is considered idle if there is no request or the component does not hold the master lease. ③ *Idle Components Recommendation.* Let $c_{\text{req}}$ denote the CPU request in the configuration, and $c_{\text{utl}}$ the most recent utilization. The allocated resources are determined as $c^* = \max(c_{\text{req}}, c_{\text{utl}})$. ④ *Load Prediction.* The recommender forecasts the load metrics for the next time frame using historical data. If the total CPU load within a time unit $\sum_i c_i' / \Delta t$ does not reach the threshold $\beta C$, indicating low likelihood of resource contention, the recommender will wait until the next period. Otherwise, it will proceed to the next steps. ⑤ *Load Calibration.* For group A components, the recommender calibrates the load using (6) with $S = 0$ to eliminate the scheduler's impact. ⑥ *Steady State Check and Recommendation.* After calibration, recommender assesses the total CPU load of busy components against the allocatable CPU resources. If the CPU load is greater, the node is in a global-unsteady state and the recommender solves (10). Otherwise, the recommender identifies local-unsteady components. It first provides recommendation for local-unsteady components, and then for steady ones. ⑦ *Memory Usage Estimation.* After recommendation,

the recommender estimates each busy component's memory usage using (3) and compares the total against available. If the available is insufficient, it will alert high risk of OOM, prompting scaling up memory to prevent potential outages.

**Weight Adjustment Mechanism.** A CPU throttling-based weight adjustment mechanism is proposed to enhance the robustness of the recommendation algorithm, given that discrepancies due to metrics inaccuracies and model deviations are inevitable.

During intervals between consecutive recommendation cycles, CPU throttling percentages $\tau$ of busy components are gathered to assess the efficacy of the previous recommendations. Ideally, these percentages should exhibit minimal variance. Deviations from this ideal trigger the following corrective actions:

① *Bottleneck Mitigation.* Components with $\tau \geq \alpha_h$ are flagged as bottlenecks, whereas those with $\tau \leq \alpha_l$ are deemed over-provisioned. To swiftly mitigate bottlenecks, over-provisioned components allocate a fraction of their CPU, $\theta c^*$, to the bottlenecks, weighted by $\tau$, ensuring a more equitable distribution.

② *Weight Adjustment.* To enhance the precision of the recommendation algorithm, the weights of bottleneck components are augmented, thereby prioritizing them for a larger allocation in subsequent cycles, which helps to counteract potential imbalances.

This weight adjustment mechanism acts as a dynamic feedback loop, responding in real-time to the system's performance fluctuations and ensuring that resource allocation remains optimally tuned amidst variability and unpredictability.

## 4.5 Updater

Upon receiving a recommendation, the updater performs a preliminary validation against a set of established update rules:

① **Alignment with Kubernetes API Specifications.** The recommendation must conform to API stipulations, including that resource allocation limits must not fall below the requested values, and that the maximum concurrency and queue length are positive.

② **Consistency in Resource Allocation.** To avert unforeseen repercussions, the updater mandates that the proposed resource allocation should not significantly differ from the previous one, quantified as $|c^* - c^*_{last}| \leq vC$, where $v$ is an adjustable parameter.

Should the recommendation pass these evaluations, the updater converts them into actionable requests. For resource allocation adjustments, it implements vertical pod autoscaling by dispatching PATCH requests to update the pods' resource limits. For traffic control parameters, the updater circumvents the necessity for pod restarts by utilizing the API Priority and Fairness (APF) mechanism [14]. It then fine-tunes the traffic control by issuing PATCH requests to adjust the APF object parameters.

Moreover, in clusters with multiple master nodes, the updater supports tailored *master instance placement*, enabling users to designate nodes for the controller manager and scheduler master instances. By default, MerKury distributes these master instances across distinct nodes to reduce resource contention. The updater enforces this placement by evicting conflicting pods from inappropriate nodes, and it only does so under low load conditions to prevent performance fluctuations during component redeployment.

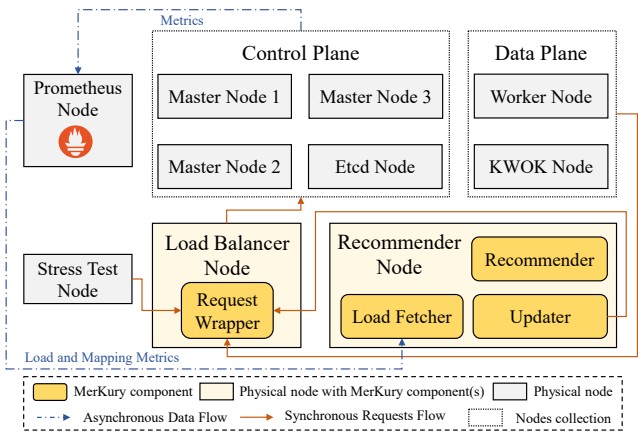

**Figure 8: Topology of the experiment environment.**

## 5 EVALUATION

MerKury is implemented with 250 lines of Lua and 5,500 lines of Python code. It leverages Openresty [28] as a load balancer, employing Lua scripts for the request wrapper. Other components operate as standalone processes, interfacing with Kubernetes clusters via the Kubernetes API. MerKury's flexible deployment requires only accessible master node IPs and a prepared kubeconfig file.

We employ an ARMA model [26] for load metric prediction. We develop a customized evolutionary algorithm to efficiently solve (9) and (10). Specifically, the CPU granularity is proportional to the remaining CPU[2], controlling the size of search space and the solution time within the threshold.

### 5.1 Environment

We constructed a cluster with high availability across three master nodes and an external high-performance etcd node to mitigate etcd's I/O bottleneck. It governed a worker node with several business containers and numerous nodes simulated by KWOK [20]. MerKury components were deployed on dedicated nodes outside of the cluster. All nodes operated within the same LAN to minimize network latency. The setup's topology is shown in Fig. 8, with node and software configurations detailed in Table 2.

### 5.2 Experiment Settings

*5.2.1 Scenarios and Datasets.* We have crafted two scenarios using a synthesized dataset to simulate large-scale Kubernetes workloads.

**Normal-Intensity Scenario.** This scenario assesses the cluster's request processing capabilities. A stress test program runs for 10 minutes, generating a mix of CRUD requests, excluding pod creations and deletions. The simulation concludes with the termination of the stress test program.

**Heavy-Intensity Scenario.** This scenario evaluates both request processing and scheduling performance. It uses the same setup as the normal scenario but adds pod creation and deletion simulations using clusterloader2 [15]. Pods are divided into saturation and latency categories, with the former representing large-scale

---

[2]remain CPU = total CPU - total CPU load / time.

**Table 2: Configuration for MerKury evaluation.**

**(a) Node configuration**

| Node | CPU | | Memory | OS |
|------|-----|-----|--------|-----|
| Master Nodes | | E5-2630 v3, 12 cores | 30GB | CentOS 7.9 |
| Load Balancer Node | | E5-2630 v3, 8 cores | 28GB | CentOS 7.9 |
| Etcd/Prometheus Node | Intel(R) Xeon(R) | Gold 6430, 128 cores | 256GB | Ubuntu 22.04 |
| Worker Node | | Platinum 8160, 96 cores | 128GB | Ubuntu 18.04 |
| KWOK/Recommender/Stress Test Node | | Silver 4210R, 40 cores | 128GB | Ubuntu 18.04 |

**(b) Software configuration**

| Software | Version | Software | Version |
|----------|---------|----------|---------|
| Kubernetes | 1.27.6 | Prometheus | 2.42.0 |
| Openresty | 1.21.4 | KWOK | 0.4.0 |
| MySQL | 8.0.35 | Redis | 4.0.14 |
| Nginx | 1.7.9 | Python | 3.10.13 |
| Go | 1.21.5 | Lua | 5.1.4 |

deployments and the latter simulating lightweight containers. The simulation is successful if all pods are running before the timeout.

**Datasets.** In the absence of public datasets on Kubernetes API call frequencies in large-scale clusters, we synthesize a dataset. The frequency of Kubernetes API calls and the number of pods to schedule increase linearly with the number of simulated nodes.

*5.2.2 Baselines.* We compare MerKury against several baselines in both scenarios.

**Vanilla Kubernetes.** This group includes `k8s-native` with default core component arguments, and `k8s-static`, which applies static argument optimization to enhance etcd storage and increase max concurrency for other components.

**P99 Baseline.** Based on the P99 algorithm from Kubernetes VPA, we implement the `p99` baseline, modified to scale the CPU usage at the 99th percentile over the last 10 minutes by 1.2 times as the maximum usage.

**Other Heuristics.** We implement three common resource allocation methods: `tsp` using the ARMA model for CPU usage prediction, `weighted` allocating CPU resources based on component load, and `evo-alg`, an evolutionary algorithm minimizing weighted average request latency.

*5.2.3 Metrics.* In both scenarios, we measure MerKury's performance against baselines using Average *Latency of All Requests (LAR)* and P99 *Latency of Read Requests (LRR)* during the stress test, representing overall request processing capability. In heavy-intensity scenarios, we also consider the average *Scheduling Throughput of Saturation Pods (STSP)*, P99 *Startup Latency of Latency Pods (SLLP)*, and the cluster's *Node Capacity* (maximum number of simulated nodes that a simulation succeeds).

For micro-benchmarks, we evaluate MerKury's overhead by measuring the average and P99 CPU and memory usage of its processes, and the average and P99 time for recommendations and updates. Additionally, we analyze the performance impacts on business workloads, including MySQL, Redis, and Nginx.

## 5.3 Results and Discussions

*5.3.1 Comparing with Baselines.* We conducted simulations with $N$ ranging from 1,000 to 10,000 in normal-intensity scenarios, and increased the number of fake nodes in heavy-intensity scenarios until failure occurred or the count reached 10,000. The performance is illustrated in Fig. 9 and 10, respectively.

In normal-intensity scenarios, MerKury significantly outperforms baselines in request latency, with minimal differences among the baselines themselves. This disparity is attributed to the limited number of busy components in such scenarios, primarily the API

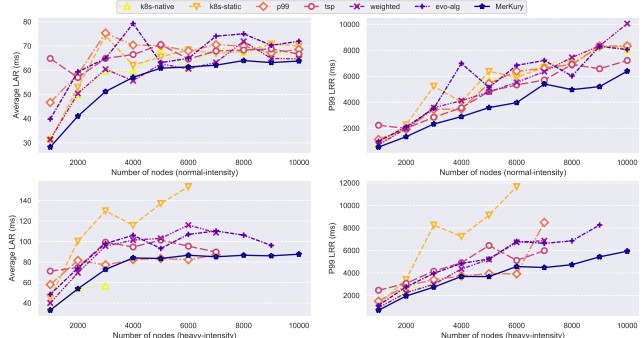

**Figure 9: Request latency.**

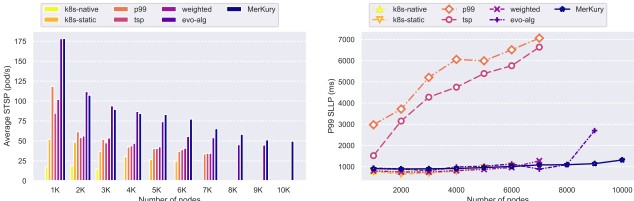

**Figure 10: Scheduling performance.**

server and controller manager, where resource contention is less of an issue, and thus, MerKury's request preprocessing provides a distinct advantage.

In heavy-intensity scenarios, MerKury achieves a node capacity of 9,000, outpacing `k8s-native` by 4.5 times. It surpasses all baselines in request latency and scheduling performance. The `p99` and `evo-alg` baselines come closest to MerKury, with `p99` providing stable API server load and lower latency, but failing to adjust effectively to the fluctuating scheduler load. `evo-alg` matches MerKury in scheduling but not in API server latency due to ignoring component dependencies, causing frequent master instance switches of the scheduler and controller-manager.

Overall, MerKury delivers substantial performance improvements, reducing average latency by 5.63% to 39.32% and P99 read latency by 15.82% to 57.73%. Its scheduling throughput is 1.14 to 16.43 times higher when managing numerous pods, and startup latency for urgent pods is reduced by up to 82.24%.



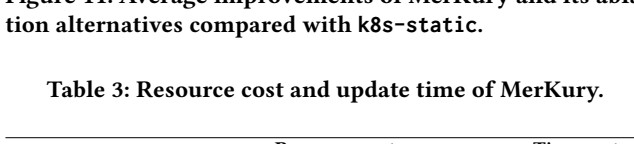

**Figure 11: Average improvements of MerKury and its ablation alternatives compared with `k8s-static`.**

**Table 3: Resource cost and update time of MerKury.**

| Scenario | Resource cost | | Time cost |
|---|---|---|---|
| | Average/P99 CPU usage (core) | Average/P99 memory usage (MB) | Average/P99 update time (s) |
| Normal-intensity | 0.01/0.14 | 162.55/163.59 | 0.23/0.57 |
| Heavy-intensity | 0.04/0.85 | 179.53/180.96 | 0.24/0.41 |

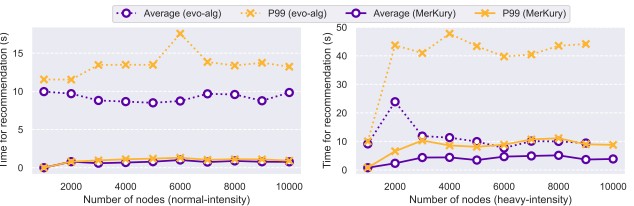

**Figure 12: Recommendation time comparison.**

*5.3.2 Ablation Study.* After proving the effectiveness of MerKury, we examine its three main components—recommendation algorithm, master instance placement, and request wrapper—to assess their individual contributions to performance.

We introduce two ablation versions of MerKury: `rec+mip` (without the request wrapper) and `rec` (further disabling master instance placement). Results for these versions are shown in Fig. 11.

The ablation study confirms the collective impact of all three components. In normal-intensity scenarios, the request wrapper significantly reduces unnecessary read request loads, particularly contributing to 73.97% of the average LAR reduction. Under heavy load, the recommendation algorithm and master instance placement play more critical roles, with contributions up to 87.7% for latency metrics and 75% for node capacity growth, respectively.

*5.3.3 Micro-benchmarks.* In micro-benchmarks, we evaluate the cost of MerKury and its impact on business workloads.

**MerKury overhead.** A monitoring thread in MerKury continuously tracks its CPU and memory usage, with time consumption for each recommendation and update recorded in logs.

As shown in Table 3, the CPU usage in normal-intensity scenarios is minimal, with the recommendation algorithm infrequently invoked. In heavy-intensity scenarios, the CPU and memory usage increase but remain within acceptable limits, with the average update time well under half a second.

Comparing the recommendation time of MerKury with `evo-alg`, as shown in Fig. 12, MerKury's average and P99 times are significantly lower, with the longest times generally within the 10-second

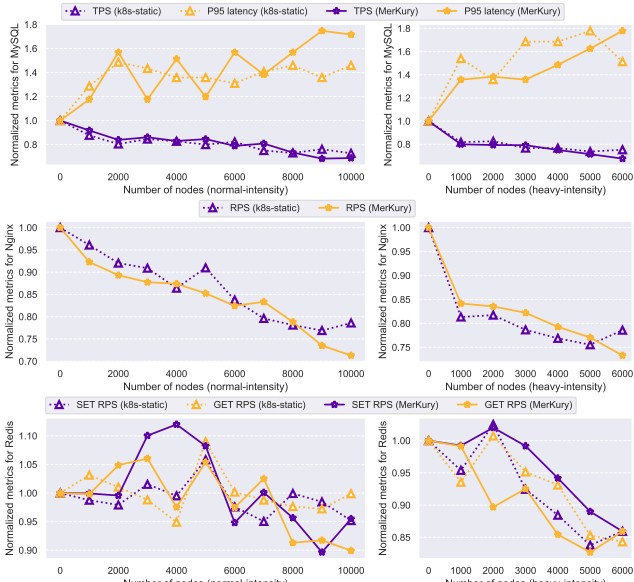

**Figure 13: Impact on business workloads (*top*: MySQL; *middle*: Nginx; *bottom*: Redis) compared with `k8s-static`.**

threshold, highlighting the efficiency of its customized evolutionary algorithm with adaptive CPU granularity.

**Impact on business workloads.** During stress tests, we also assess the performance of business workloads to determine any potential negative impact of MerKury. As depicted in Fig. 13, business workload performance degrades with increasing cluster load and node count. However, the difference between MerKury and `k8s-static` is negligible, indicating that MerKury has minimal negative impact on business workloads, with performance degradation primarily due to cluster scale and load.

## 6 CONCLUSION

In this paper we presented MerKury, a lightweight framework developed to enhance the Kubernetes performance for large-scale clusters. To address the critical issues of read request load and resource contention, we introduced a request wrapper for preprocessing requests and a recommendation algorithm alongside a master instance placement mechanism to optimize resource allocation. Comprehensive experiments have demonstrated that the framework significantly improved both request processing and scheduling throughput, showcasing its ability to bolster Kubernetes performance without adding substantial overhead.

In future endeavors, we intend to enhance MerKury's predictive capabilities using deep learning techniques, thereby refining its adaptability in the ever-changing landscape of cloud environments.

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

## A SYSTEM DETAILS

### A.1 Queuing Model Details

We adopt the following assumptions for queuing model construction to simplify the modeling so that the queuing model can be characterized as a birth-death process: ① The CPU-concurrency mapping for each component is invariant across different request types, with precedence given to mappings derived from the load fetcher. ② In scenarios where multiple requests are concurrently executed, each request is allocated an equitable share of CPU resources. ③ The inter-arrival time of requests adheres to an exponential distribution with rate $n_r/\Delta t$. ④ The execution time conforms to an exponential distribution with rate CPU Utilization/Request CPU Load.

In the steady state, the rate of transitions away from a state equals the rate of transitions into that state, leading to the global

balance equations:

$$\begin{cases} p_0 \lambda_0 = p_1 \mu_1, \\ p_{f^*+q^*-1} \lambda_{f^*+q^*-1} = p_{f^*+q^*} \mu_{f^*+q^*}, \\ p_{n-1} \lambda_{n-1} + p_{n+1} \mu_{n+1} = p_n (\lambda_n + \mu_n). (1 \le n < f^* + q^*) \end{cases} \quad (11)$$

Given that the sum of probabilities must equal 1, we integrate this constraint into the global balance equations to solve for the probability distribution $\{p_n\}$ as:

$$p_n = p_0 \prod_{i=1}^{n} \frac{\lambda_{i-1}}{\mu_i} \quad (1 \le n \le f^* + q^*),$$

$$p_0 = \left( 1 + \sum_{n=1}^{f^*+q^*} \prod_{i=1}^{n} \frac{\lambda_{i-1}}{\mu_i} \right)^{-1}. \quad (12)$$

The expected values for $L$, $L_q$ and $L_e$ are derived as:

$$L = \sum_{n=1}^{f^*+q^*} n p_n, \quad L_q = \sum_{n=f^*+1}^{f^*+q^*} (n - f^*) p_n,$$

$$L_e = L - L_q = \sum_{n=1}^{f^*} n p_n + \sum_{n=f^*+1}^{f^*+q^*} f^* p_n. \quad (13)$$

Considering the finite capacity of the queuing system, from the servers' perspective, the effective arrival rate $\lambda_{\text{eff}} = \lambda(1 - p_{\text{reject}})$. Employing Little's Law, $W$, $W_q$ and $W_e$ are calculated accordingly:

$$W = L/\lambda_{\text{eff}}, \quad W_q = L_q/\lambda_{\text{eff}}, \quad W_e = L_e/\lambda_{\text{eff}}. \quad (14)$$

In the unsteady state, where the server operates at full capacity, the average number of executing requests, $L_e$, is estimated to be $f^*$. Concurrently, $L_q$ is deduced from the number of requests that correspond to the unprocessed CPU load:

$$L_q = l n_r / c. \quad (15)$$

## A.2 Monotonicity Discussions

The subsequent discussions presume that the queuing system has reached a steady state. For the sake of brevity, we define $a_n = \prod_{i=1}^{n} \lambda_{i-1}/\mu_i$.

**THEOREM A.1.** *The average latency $W$ exhibits a monotonic decrease with the CPU allocation $c^*$ for $c^* \le \max f(x)$, with $\max f(x)$ representing the peak CPU utilization.*

**PROOF.** Let $0 < c_1^* < c_2^* \le \max f(x)$, and denote $\mathcal{P} = p_0(c_1^*)/p_0(c_2^*)$, $\mathcal{A}_n = a_n(c_1^*)/a_n(c_2^*)$. The relationship is given by:

$$\frac{p_n(c_1^*)}{p_n(c_2^*)} = \frac{a_n(c_1^*) p_0(c_1^*)}{a_n(c_2^*) p_0(c_2^*)} = \mathcal{P} \mathcal{A}_n, \quad \frac{\mathcal{A}_n}{\mathcal{A}_{n-1}} = \frac{\mu_n(c_2^*)}{\mu_n(c_1^*)}. \quad (16)$$

To prove the above theorem is equivalent to proving that $W(c_1^*) > W(c_2^*)$.

Initially, we discuss the probability distribution. In accordance with Equation (1), the service rate $\mu_n$ increases monotonically with $c^*$ for $c^* \le \max f(x)$. Given that the arrival rate $\lambda_n$ is independent of $c^*$, it follows that $a_n$ decreases monotonically with $c^*$. Consequently, the probability of the system being idle $p_0$ escalates with $c^*$. We identify $x_1$ as the smallest x-coordinate where the curves $y = f(\min(x, f^*))$ and $y = c_1^*$ as $x_1$ intersect. For $n \le \lfloor x_1 \rfloor$, $\mu_n(c_1^*) = \mu_n(c_2^*)$, thus $\mathcal{A}_n = 1$. Since $\mathcal{P} < 1$, it is

clear that $p_n(c_1^*) < p_n(c_2^*)$. For $n \ge \lceil x_1 \rceil$, $\mu_n(c_1^*) < \mu_n(c_2^*)$, and therefore $\mathcal{A}_n/\mathcal{A}_{n-1} > 1$, indicating an increase in $A_n$ with $n$. Given that $\sum_{n=0}^{f^*+q^*} p_n(c_1^*) = \sum_{n=0}^{f^*+q^*} p_n(c_2^*) = 1$, there exists $n_1 \in [\lceil x_1 \rceil, f^* + q^*]$ such that $\mathcal{A}_{n_1} \le 1/\mathcal{P} \le \mathcal{A}_{n_1+1}$. For $n \le n_1$, $p_n(c_1^*) < p_n(c_2^*)$; whereas for $n \ge n_1 + 1$, $p_n(c_1^*) > p_n(c_2^*)$. Hence $p_{\text{reject}}(c_1^*) > p_{\text{reject}}(c_2^*)$. Additionally, the following equation holds:

$$\sum_{n=0}^{n_1} p_n(c_2^*) - p_n(c_1^*) = \sum_{n=n_1+1}^{f^*+q^*} p_n(c_1^*) - p_n(c_2^*). \quad (17)$$

Subsequently, we discuss the monotonicity of the average number of requests in the system, $L$, with respect to $c^*$. The calculation is as follows:

$$\begin{aligned} L(c_1^*) - L(c_2^*) &= \sum_{n=1}^{f^*+q^*} n \left( p_n(c_1^*) - p_n(c_2^*) \right) \\ &= -\sum_{n=0}^{n_1} n \left( p_n(c_2^*) - p_n(c_1^*) \right) + \\ &\quad \sum_{n=n_1+1}^{f^*+q^*} n \left( p_n(c_1^*) - p_n(c_2^*) \right) \\ &> -n_1 \sum_{n=0}^{n_1} p_n(c_2^*) - p_n(c_1^*) + \\ &\quad (n_1 + 1) \sum_{n=n_1+1}^{f^*+q^*} p_n(c_1^*) - p_n(c_2^*) \\ &= \sum_{n=0}^{n_1} p_n(c_2^*) - p_n(c_1^*) > 0. \end{aligned} \quad (18)$$

Thus, $L$ decreases monotonically with $c^*$. Since $W = L/\lambda(1 - p_{\text{reject}})$ and $p_{\text{reject}}(c_1^*) > p_{\text{reject}}(c_2^*)$, it follows that $W(c_1^*) > W(c_2^*)$. $\square$

**THEOREM A.2.** *The average request number $L$ increases monotonically with respect to the maximum concurrency $f^*$ if increasing $f^*$ does not enhance the service rate.*

**PROOF.** Given that $f^*$ is a natural number, the proof hinges on proving $L(f^*) < L(f^*+1)$, under the condition that $\min(c^*, f(f^*))$ is greater than or equal to $\min(c^*, f(f^*+1))$. We proceed to examine the scenarios separately.

When $\min(c^*, f(f^*)) > \min(c^*, f(f^* + 1))$, the relationships between $\mu_n$ and $a_n$ for $f^*$ and $f^* + 1$ are:

$$\begin{aligned} \mu_n(f^*) = \mu_n(f^* + 1) \quad a_n(f^*) = a_n(f^* + 1) \quad (1 \le n \le f^*), \\ \mu_n(f^*) > \mu_n(f^* + 1) \quad a_n(f^*) < a_n(f^* + 1) \quad (n \ge f^* + 1). \end{aligned} \quad (19)$$

According to Equation (12), it is inferred that $p_0(f^*) > p_0(f^* + 1)$. Drawing parallels to the discussions on the probability distribution for varying $c^*$, there exists $n_1 \in [f^*, f^* + q^*]$ such that for $0 \le n \le n_1$, $p_n(f^*) > p_n(f^* + 1)$; and for $n \ge n_1 + 1$, $p_n(f^*) \le p_n(f^* + 1)$. Considering that the sum of probabilities equals one, we derive that

$$\sum_{n=0}^{n_1} p_n(f^*) - p_n(f^* + 1) = \sum_{n=n_1+1}^{f^*+q^*+1} p_n(f^* + 1) - p_n(f^*). \quad (20)$$

Subsequently, we compare $L(f^*)$ and $L(f^*+1)$ through subtraction:

$$L(f^*) - L(f^* + 1) = \sum_{n=1}^{f^*+q^*} n p_n(f^*) - \sum_{n=1}^{f^*+q^*+1} n p_n(f^* + 1)$$

$$= \sum_{n=0}^{n_1} n(p_n(f^*) - p_n(f^* + 1)) -$$

$$\sum_{n=n_1+1}^{f^*+q^*+1} n(p_n(f^* + 1) - p_n(f^*))$$

$$< n_1 \sum_{n=0}^{n_1} p_n(f^*) - p_n(f^* + 1) -$$

$$(n_1 + 1) \sum_{n=n_1+1}^{f^*+q^*+1} p_n(f^* + 1) - p_n(f^*)$$

$$= - \sum_{n=0}^{n_1} p_n(f^*) - p_n(f^* + 1) < 0. \quad (21)$$

When $\min(c^*, f(f^*)) > \min(c^*, f(f^* + 1))$, altering the maximum concurrency from $f^*$ to $f^* + 1$ introduces no change for $\mu_n$ and $a_n$ but increases the number of terms in the summation of Equation (12). Consequently, $p_0(f^* + 1) < p_0(f^*)$. Let $\Delta p_0 = p_0(f^*) - p_0(f^* + 1)$, and given the sum of probabilities is unit, it follows that

$$p_{f^*+q^*+1}(f^* + 1) = \sum_{n=0}^{f^*+q^*} p_n(f^*) - p_n(f^* + 1)$$

$$= \left(1 + \sum_{n=1}^{f^*+q^*} a_n\right) \Delta p_0. \quad (22)$$

We then compare $L(f^*)$ and $L(f^* + 1)$ via subtraction:

$$L(f^*) - L(f^* + 1) = \sum_{n=1}^{f^*+q^*} n \left(p_n(f^*) - p_n(f^* + 1)\right) -$$

$$(f^* + q^* + 1) p_{f^*+q^*+1}(f^* + 1)$$

$$< \Delta p_0 (f^* + q^* + 1) \sum_{n=1}^{f^*+q^*} a_n - \quad (23)$$

$$\Delta p_0 (f^* + q^* + 1) \left(1 + \sum_{n=1}^{f^*+q^*} a_n\right)$$

$$= - \Delta p_0 (f^* + q^* + 1) < 0.$$

Hence, $L(f^*) < L(f^* + 1)$ for $\min(c^*, f(f^*)) \geq \min(c^*, f(f^* + 1))$. □

Theorem A.3. *Regarding the maximum queue length $q^*$, the probability of request rejection, $p_{reject}$, decreases monotonically. Concurrently, the average number of requests in the system $L$, the average number in the queue $L_q$, and the average number in execution $L_e$, all increase monotonically.*

Proof. Similar to $f^*$, $q^*$ is a natural number, and the proof of monotonicity for a function $g(q^*)$ with respect to $q^*$ is equivalent to comparing $g(q^*)$ with $g(q^* + 1)$.

Adjusting $q^*$ does not affect $\lambda_n$, $\mu_n$, or $a_n$, but it does increase the number of terms in the summation for calculating $p_0$. Given that $\lambda_n$ and $\mu_n$ are non-negative, $p_0$ decreases monotonically with $q^*$. In the expression for $p_n$, only $p_0$ depends on $q^*$, hence $p_n$ also decreases monotonically with $q^*$. Let $\Delta p_0 = p_0(q^*) - p_0(q^* + 1)$, it follows that $p_n(q^*) - p_n(q^* + 1) = \Delta p_0 a_n$ for $n \leq f^* + q^*$. Considering the sum of probabilities is unity, we have

$$p_{f^*+q^*+1}(q^* + 1) = \Delta p_0 \left(1 + \sum_{n=1}^{f^*+q^*} a_n\right). \quad (24)$$

We compare $p_{\text{reject}}(q^*)$ and $p_{\text{reject}}(q^* + 1)$ via subtraction,

$$p_{\text{reject}}(q^*) - p_{\text{reject}}(q^* + 1)$$

$$= a_{f^*+q^*} p_0(q^*) - a_{f^*+q^*+1} p_0(q^* + 1)$$

$$= a_{f^*+q^*} \left(p_0(q^*) - \frac{\lambda_{f^*+q^*}}{\mu_{f^*+q^*+1}} p_0(q^* + 1)\right). \quad (25)$$

Given the system is in steady state, $0 < \frac{\lambda_{f^*+q^*}}{\mu_{f^*+q^*+1}} < 1$. Since $p_0(q^*) > p_0(q^* + 1)$, it follows that $p_{\text{reject}}(q^*) > p_{\text{reject}}(q^* + 1)$.

Subsequently, we compare $L$, $L_q$, and $L_e$ using the same method:

$$L(q^*) - L(q^* + 1) = \sum_{n=1}^{f^*+q^*} n(p_n(q^*) - p_n(q^* + 1)) -$$

$$(f^* + q^* + 1) p_{f^*+q^*+1}(q^* + 1)$$

$$= \Delta p_0 \sum_{n=1}^{f^*+q^*} n a_n - \quad (26)$$

$$\Delta p_0 (f^* + q^* + 1) \left(1 + \sum_{n=1}^{f^*+q^*} a_n\right)$$

$$< - \Delta p_0 (f^* + q^* + 1) < 0,$$

$$L_q(q^*) - L_q(q^* + 1) = \Delta p_0 \sum_{n=f^*+1}^{f^*+q^*} (n - f^*) a_n -$$

$$\Delta p_0 (q^* + 1) \left(1 + \sum_{n=1}^{f^*+q^*} a_n\right)$$

$$< \Delta p_0 (q^* + 1) \sum_{n=f^*+1}^{f^*+q^*} a_n - \quad (27)$$

$$\Delta p_0 (q^* + 1) \left(1 + \sum_{n=1}^{f^*+q^*} a_n\right)$$

$$< - \Delta p_0 (q^* + 1) < 0,$$

$$L_e(q^*) - L_e(q^* + 1) = \Delta p_0 \left(\sum_{n=1}^{f^*} n a_n + f^* \sum_{n=f^*+1}^{f^*+q^*} a_n\right) -$$

$$f^* p_{f^*+q^*+1}(q^* + 1)$$

$$< \Delta p_0 f^* \sum_{n=1}^{f^*+q^*} a_n - \Delta p_0 f^* \left(1 + \sum_{n=1}^{f^*+q^*} a_n\right) \quad (28)$$

$$< - \Delta p_0 f^* < 0.$$

Therefore, $L$, $L_q$, and $L_e$ increase monotonically with respect to $q^*$.                                                                                      □

## A.3 Traffic Control Parameters Updating

For traffic control parameters recommendation, instead of changing arguments that require pod restart, the updater adopts the API priority and fairness (APF) mechanism in which requests are classified into multiple priority groups and each group occupies different seats. Specifically, the maximum concurrency of the API server is set as $F$ which is large enough (e.g. 3,000), and then a `PriorityLevelConfiguration` (PLC) object named "merkury-empty" is created in the cluster. "Merkury-empty" doesn't match any request and it becomes a "placeholder". The more "seats" it occupies, the lower the effective maximum concurrency of the API server. Therefore, the updater changes $f^*$ by sending `PATCH` requests to update the `nominalConcurrencyShares` value of "merkury-empty". For $q^*$ changes, it sends `PATCH` requests to update `queueLengthLimit` values of Kubernetes' default PLCs.

It is important to acknowledge that the APF mechanism, which incorporates multiple queues, diverges somewhat from the idealized single-queue model we have formulated. Nonetheless, the efficacy of the traffic control parameter recommendation strategy has been validated. Moreover, considering that only the API server possesses configurable traffic control parameters, the recommender confines its traffic control parameter recommendations to the API server, with the updater making corresponding updates exclusively to the API server's traffic control parameters.

## B EXPERIMENT ADDENDUM

### B.1 Detailed Settings

**Parameter Settings.** The minimum availability $P$ is set at 99.9%. The CPU resource contention threshold $\beta$ is 0.7. Initial weight parameters are uniformly set to 1, incrementing to $w' = w + 1$ during adjustments. For group A components, bottleneck and over-allocation thresholds are $\alpha_h = 0.6$ and $\alpha_l = 0.3$; other components use 0.7 and 0.4, respectively. The reallocation fraction $\theta$ is 0.1, and the maximum allocation deviation fraction $v$ is 0.2. The recommendation time threshold $T$ is capped at 10 seconds. The recommendation period $\Delta t$ is set to 1 minute, balancing the collection of valid incremental metrics with timely recommendation updates.

For the evolution algorithm, the population size is established at 1,000, with a maximum of 100 generations. The probabilities for selection, crossover, and mutation are set at 0.2, 0.8, and 0.05, respectively. Within the early stopping strategy, if the variation in fitness values over 10 successive generations does not exceed 0.01, the algorithm is deemed to have converged, prompting the optimizer to terminate.

**Heavy-Intensity Scenario.** In heavy-intensity scenario, clusterloader2 is used to simulate pod creations and deletions. Specifically, pods are categorized into saturation pods and latency pods. Saturation pods are managed by deployments with 3,000 replicas to simulate large-scale workloads deployed on a large number of nodes, while latency pods are controlled by deployments with only one replica to simulate lightweight containers. Pod creation and deletion start simultaneously with the stress test program. In a

**Table 4: Argument optimizations for baselines except for `k8s-native`.**

| Component | Arguments |
|---|---|
| etcd | –quota-backend-bytes=8589934592 |
| API server | –max-requests-inflight=2000
–max-mutating-requests-inflight=1000 |
| controller manager | –kube-api-qps=200
–kube-api-burst=300 |
| scheduler | –kube-api-qps=200
–kube-api-burst=400 |

simulation with $N$ simulated nodes, $30N$ saturation pods are created in their entirety first, and once they are all running or until a saturation timeout of 100 minutes is reached, the creation for $N$ latency pods starts at a rate of RPS equal to 5. Once all the latency pods are running or until a latency timeout of 15 minutes from the dispatch of the last latency pod creation request is reached, all the pods are finally deleted. The simulation ends with the finish of both the stress test program and the pod creation and deletion process. The simulation fails when any timeout is reached.

**Datasets.** To simulate a mix of CRUD requests using the stress test program, we synthesize a dataset in which there are 18 types of common objects, and the fraction of calls with different verbs are estimated as

PUT : DELETE : PATCH : POST : LIST : GET = 1 : 1 : 2 : 4 : 6 : 8,

based on the Pinterest's statistics [9]. For the same verb, the fractions of calls with different objects are equal to each other. Besides, it is assumed that the frequency grows linearly with the fake node number. For example, when there are 1,000 simulated nodes, the RPS of PUT endpoint requests is set to 1, and accordingly when there are 2,000 simulated nodes, the RPS of POST secret requests equals 8.

**Static Arguments Optimization.** During the evaluation of vanilla Kubernetes, we have found that the default parameters of control plane components are suboptimal in large-scale clusters. Specifically, the etcd storage is insufficient and the concurrency of other components are severely constrained. Therefore, for baselines except for `k8s-native`, we have applied static arguments optimization as shown in table 4.

