# OpenReview forum: "Mer\underline{K}ury: Adaptive Resource Allocation to Enhance the \underline{K}ubernetes Performance for Large-Scale Clusters"
_ACM.org/TheWebConf/2025/Conference — WWW 2025 Poster_

### Official Review · Reviewer_CcRS · 2024-11-15

**Novelty:** 4
**Technical Quality:** 3

**Review:**

Summary of the work
This paper presents MerKury to enhance the performance of Kubernetes master node, mainly for API server access latency and scheduling throughput. The proposed system is well presented using distinct components. The request wrapper is responsible to reduce API server latency by adding resourceVersion header and splitting a response to fit in a memory.  The recommender is responsible for allocating CPU and memory resources to a control module to maximize scheduling throughput. The authors used queueing theory for recommendations. The system is evaluated in a simulated environment using KWOK, which seems to be reasonable approach considering the scale that the proposed system is targeting. The proposed system shows superb performance compared to other baseline implementation.

Clarity
The authors clearly states its goals and the mechanisms to achieve the goal. However, the explanation and the data source of motivating examples (Figure 1 and Figure 5) are not clear. (Question 1,2,3)

Originality and significance
As far as I know, applying queueing theory to decide optimal compute resource allocation is new. However, for the given problem domain of enhancing the master node performance, the proposed algorithm seems to be over-engineered. Details in Questions 4.

Questions.
1. The request wrapper adds "resourceVersion" parameter when it is missing to avoid unnecessary access to etcd.  https://kubernetes.io/docs/reference/using-api/api-concepts/#efficient-detection-of-changes Referencing this page, it takes a role of expressing version. Is there any unexpected result by setting the value as resourceVersion=0 ?
2. It was difficult to understand the intuition behind Figure 5. Why the theoritical mapping is capped at 12 (CPU utilization percent?), and why the average CPU utilization is lower? Is it  because the additional overheads from I/O? This is closely related to local-unsteady state of lower CPU utilization but difficult to understand.
3. How did the authors get data which is drawn in Figure 1? Is it from a real world deployment? Additional explanation would be nice.
4. Theoritical explanation using queueing theory for the recommender module is nice. However, for me, it seems over-engineered. As far as I understand, it mutually sets up CPU and memory configurations of control plane to improve overall scheduling throughput. In general, as most of other large-scale systems, such as Hadoop and HDFS, the main node is crucial to run the system properly. Because of this, administrator takes great care of main nodes with enough resources even sacrificing worker nodes. I am wondering how the performance of MerKury is compared to a case where a master node has enough CPU and memory resources. Assuming that there are over 1000s of worker nodes, administrators are not likely to squeeze the master node resource allocation to run the entire system reliably. Most likely, they will sacrifice the work nodes to provide more resources to a master node.
5. It would be nice if the authors could compare with a simple reactive algorithm without the complex procedure in the Recommender module. As a system administrator role, one is likely to allocate simply more resources as the system becomes unsteady state. Is it similar to one in the other heuristics in the experiment section? If so, what is the intuition behind the MerKury recommender performs better than the others? Adding more practical reason, other than the algorithmic explanation, would be nice.
6. How did the authors evaluate the API server (regarding the request wrapper)? The proposed system is beneficial only when "resourceVersion" parameter is missing, and when the output size is very large. Did all the requests in the experiments satisfy this condition? I understand the limitation of lacking real-world K8s workload scenario, but the evaluation results and performance can differ greatly based on the request scenario.
7. Detailed explanation and intuition behind the business workload evaluations are necessary.
8. What is the weighted average request latency? how is it different from one without weighting?

**Questions:**

1. The request wrapper adds "resourceVersion" parameter when it is missing to avoid unnecessary access to etcd.  https://kubernetes.io/docs/reference/using-api/api-concepts/#efficient-detection-of-changes Referencing this page, it takes a role of expressing version. Is there any unexpected result by setting the value as resourceVersion=0 ?
2. It was difficult to understand the intuition behind Figure 5. Why the theoritical mapping is capped at 12 (CPU utilization percent?), and why the average CPU utilization is lower? Is it  because the additional overheads from I/O? This is closely related to local-unsteady state of lower CPU utilization but difficult to understand.
3. How did the authors get data which is drawn in Figure 1? Is it from a real world deployment? Additional explanation would be nice.
4. Theoritical explanation using queueing theory for the recommender module is nice. However, for me, it seems over-engineered. As far as I understand, it mutually sets up CPU and memory configurations of control plane to improve overall scheduling throughput. In general, as most of other large-scale systems, such as Hadoop and HDFS, the main node is crucial to run the system properly. Because of this, administrator takes great care of main nodes with enough resources even sacrificing worker nodes. I am wondering how the performance of MerKury is compared to a case where a master node has enough CPU and memory resources. Assuming that there are over 1000s of worker nodes, administrators are not likely to squeeze the master node resource allocation to run the entire system reliably. Most likely, they will sacrifice the work nodes to provide more resources to a master node.
5. It would be nice if the authors could compare with a simple reactive algorithm without the complex procedure in the Recommender module. As a system administrator role, one is likely to allocate simply more resources as the system becomes unsteady state. Is it similar to one in the other heuristics in the experiment section? If so, what is the intuition behind the MerKury recommender performs better than the others? Adding more practical reason, other than the algorithmic explanation, would be nice.
6. How did the authors evaluate the API server (regarding the request wrapper)? The proposed system is beneficial only when "resourceVersion" parameter is missing, and when the output size is very large. Did all the requests in the experiments satisfy this condition? I understand the limitation of lacking real-world K8s workload scenario, but the evaluation results and performance can differ greatly based on the request scenario.
7. Detailed explanation and intuition behind the business workload evaluations are necessary.
8. What is the weighted average request latency? how is it different from one without weighting?

**Reviewer Confidence:**

3: The reviewer is confident but not certain that the evaluation is correct

**Scope:**

3: The work is somewhat relevant to the Web and to the track, and is of narrow interest to a sub-community

---

### Official Review · Reviewer_q24H · 2024-11-17

**Novelty:** 3
**Technical Quality:** 5

**Review:**

This paper presents MerKury, a lightweight framework to improve scheduling throughput and resource allocation for large-scale k8s. There are two main contributions of this paper, including the pre-processing for alleviating CPU load, and an adaptive resource allocation to mitigate resource contention.

Overall, this is a solid work. However, there exist some motivation (novelty) issues and technique issues. If authors can clarify the following concerns, I would happy to change ratings.

**Questions:**

1. Where is the data in figure 1 collected from? Authors should cite the resource.

2. Why do authors mention multi-cluster strategies and single-cluster strategies? What is the relationship between MerKury and two strategies? Is MerKury actually an advancement of one of the two strategies? Since authors claim that multi-cluster strategies have disadvantages like additional complexity and resource overheads, I assume that MerKury should be a single-cluster strategy.

3. Without altering the Kubernetes codebase may not be a good motivation. If a lightweight alteration do not incur any insufficiency, how do authors distinguish MerKury from Kubernetes optimization works?

4. In addition, I hate to say this, but resource allocation solutions for k8s have been too many. The section 3.2 should be more detailed for convincing. Current version is shallow, which cannot guarantee the novelty.

5. Since authors claim that M/M/1 and M/M/c is not practical, why choose the average inter-arrival time inverse as the arrival rate?

6. I am a little confused about Eq. (9). What are the variables that need to be determined? Authors can give a notation list about parameters and variables.

7. Since authors choose the heuristic algorithm for steady state problem, the approximation may not be guaranteed.

Other suggestions:
The font size in almost every figure, especially in experimental result figures, is extremely small. I can barely see sharp. Please do resize your font in all figures.

**Reviewer Confidence:**

3: The reviewer is confident but not certain that the evaluation is correct

**Scope:**

3: The work is somewhat relevant to the Web and to the track, and is of narrow interest to a sub-community

---

### Official Review · Reviewer_wATt · 2024-11-28

**Novelty:** 5
**Technical Quality:** 6

**Review:**

## Summary
This paper presents MerKury, a lightweight framework designed to enhance the performance of Kubernetes in large-scale clusters. The framework addresses the scalability issue by employing a dual strategy: preprocessing specific requests to alleviate unnecessary load and introducing an adaptive resource allocation algorithm to mitigate resource contention. The authors evaluate MerKury across various cluster scales and demonstrate significant improvements over vanilla Kubernetes, achieving up to 16.4x higher scheduling throughput and reducing request latency by up to 39.3%.

## Strengths
1. The proposed framework is reasonable. The authors provide clear evidence of the challenges and then provide solutions accordingly.
2. The framework serves as a non-intrusive plugin, which ensures broad applicability and ease of deployment.
3. Empirical results show clear improvements over vanilla and heuristic baselines.

## Weakness
1. The writing is quite hard to follow. The authors may provide more background information for terms related tu Kubernetes.
2. The results on business workloads are not significant.

**Questions:**

## Questions
1. The results on business workloads seem not significant. For example, on Nginx workload, k8s-static performs better than the proposed method when the number of nodes are 10000 for normal-intensity and 6000 for heavy-intensity. In addition, the results fluctrate on Redis benchmark. Why does the proposed method perform not as significant as on synthetic workloads? More analysis can be provided.

## Minors
1. Figure 1 should be provided in pdf format. The font size in most figures is too small to read, e.g., Figure 2, Figures 9-13.
2. Section 4.3.2, "previous" -> "Previous".

**Reviewer Confidence:**

1: The reviewer's evaluation is an educated guess

**Scope:**

3: The work is somewhat relevant to the Web and to the track, and is of narrow interest to a sub-community

---

### Official Review · Reviewer_nZwz · 2024-12-01

**Novelty:** 4
**Technical Quality:** 4

**Review:**

From my extremely limited expertise in this area, this paper seems quite elaborate and mathematically sound in its design of MerKury which in my understanding is a recommender system to enhance the performance Kubernetes on large-scale clusters. I have two comments/questions regarding the current version of the draft.

1. Can the cap limit on the size of the objects returned have some adverse effects? For example, can there be some applications which may need to perform LIST operations (like: "LIST all pods") and if there are how will they be managed? Does this limit means that only a certain set of records (here the pods) will be retrieved and not more than that or will there be multiple subsequent responses that would contain the remaining list? I guess this portion was not clear to me.

2. I would suggest that the authors make the captions of the figures in the evaluation section a bit more elaborate highlighting what is the key insight that one should look for in those figures. This would help in increasing the readability of the draft. Also, the legends, x-ticks, y-ticks were not visible. The authors may take a print copy of the paper to ensure whether they can see what is written in the figures without any magnification.

3. Other than this I am unsure how this paper is compared with any other state-of-the-art baseline (other than a few in-house baselines), if there are any.

**Questions:**

Please see the main review for the same.

**Reviewer Confidence:**

2: The reviewer is willing to defend the evaluation, but it is likely that the reviewer did not understand parts of the paper

**Scope:**

3: The work is somewhat relevant to the Web and to the track, and is of narrow interest to a sub-community

---

### Official Review · Reviewer_Hs9J · 2024-12-02

**Novelty:** 4
**Technical Quality:** 5

**Review:**

Strengths:

S1. The problem of resource allocation is of great impact in real work applications, and directly relevant to the industrial deployment.

S2. The lightweight non-instrusive design of the proposed method is very useful.

S3. The improvements over the original baselines are large.

Weakness:

W1. The paper is not easy to parse for the researchers who haven’t done some in-depth development using Kubernets, even with the presence of Figure 2 and Figure 3. Some end-to-end examples, i.e., what one specific Kubernetes request looks like and how it is processed without MerKury and with MerKury, would be very useful.

W2. The queueing model seems to be a core design in MerKury. At the same time, the queueing problem is a classic problem in the network and resource allocation literature. It would be useful if the authors can clarify the technical innovation of the proposed model (in terms of previous models cannot be simply adjusted and adopted), and conduct comparison experiments if possible.

W3. ARMA model is employed to predict load metric. However, load metric prediction, as in a broader context of time series prediction, is by itself an actively studied problem. ARMA seems to be a classic model that has been outperformed by many modern methods. I would suggest the authors to justify the choice of ARMA, i.e., whether and why it is good enough.

W4. The authors have discussed a few alternative resource allocation methods in Section 3.2. Although they are intrusive, it would still be beneficial to compare MerKury with them, or clearly discuss why they cannot be included in the experimental evaluation, instead of the basic baselines in Section 5.2.2.

Revisions:

R1. The notations in Figure 2 are not visible, as well as those in Figure 9, 10, 11, 12, 13. They should have the same font size as the main text.

R2. The first paragraph in Section 4.1 repeats the intuitions in the introduction without any additional details. I would suggest to shorten or remove it.

R3. line 367, “previous studies …” should be capitalized.

**Questions:**

Please discuss the concerns and suggestions listed in W1-W4 and R1-R3.

**Reviewer Confidence:**

3: The reviewer is confident but not certain that the evaluation is correct

**Scope:**

3: The work is somewhat relevant to the Web and to the track, and is of narrow interest to a sub-community